# Analysis of influence of filling body with different filling intervals on stope stability

**Chunming Ai**[1,2], **Anju Yu**[1,2]*, **Li Zhang**[3], **Yingbo Wen**[1,2]

1 College of Safety Science and Engineering, Liaoning Technical University, Huludao, 125000, China, 2 Key Laboratory of Thermal Disaster and Prevention, Ministry of Education, Huludao, 125000, China, 3 Shanxi Jinshen Energy Co., Ltd., Xinzhou, 034000, China

* 849119477@qq.com

**Data Availability Statement:** All relevant data are within the paper.

**Funding:** The work was financially supported by The Natural Science Foundation Program of Liaoning Province(No.2022-MS-395).All authors

## Abstract

The stratified backfill will lead to the reduction of the strength of the filling body, thus increasing the safety risks of the stope. By means of experimental research and numerical simulation, the stability of stope is studied. Based on the physical parameters of filling body measured by triaxial compression test, the stability of filling stope is numerically simulated by FLAC 3D software. By comparing the stress distribution and displacement of stope with filling body at different intervals, the filling process parameters of Bianjiadayuan lead-zn-silver mine can be optimized.The experimental results show that: When the filling interval is 12h, the roof stress, filling body pressure stress and roof displacement are small, and the stope stability is better. When it exceeds 12h, the change is large and the stope stability is poor. The interval time in the stratified backfill process of Bianjiadayuan lead-zinc-silver mine should not exceed 12h.The research results have significant reference value for improving the stope stability and selecting a reasonable interval time for stratified backfill.

## Introduction

The ideal cemented paste backfill(CPB) is to use slurry to fill the mined-out areas at one time, but at present, most mines use the stratified backfill [1,2]. There are three main reasons for the occurrence of stratified backfill: Firstly, the mined-out areas volume of domestic metal mines generally reaches tens of thousands to hundreds of thousands of cubic meters, while the filling capacity of the filling system is 60~150m³/h, and the continuous operation time of the filling system is about 8h~12h [3]. Therefore, in the actual mined-out areas filling, the filling capacity of the filling station cannot meet the filling demand of large volume mined-out areas, resulting in the phenomenon of multiple filling [4,5]; Second, the height of some filling spaces is too large.If a large amount of filling slurry is filled at one time, the filling retaining wall will not be able to bear the hydrostatic pressure brought by the slurry [6]. In order to avoid the collapse of filling retaining wall and other safety problems, the stratified backfill method is often adopted in the actual production, that is, the secondary filling is carried out after the initial filling slurry is condensed [7,8]. This method can effectively reduce the pressure of filling retaining wall; Third, filling pipeline blockage or filling equipment failure will also lead to the suspension of filling operations [9–11].

are involved in the study design, data collection and analysis, decision to publish, and preparation of the manuscript.

**Competing interests:** The authors have declared that no competing interests exist.

To solve the stability problem of filling stope, numerical simulation method is widely used worldwide, which can directly reflect the change characteristics of displacement of detection points. Zhao [12] used FLAC 3D for modeling and Mohr-Coulomb yield criterion to detect detection points in the filling region and observe their displacement. Chen [13] proposed 3DMine-Flac3D coupling modeling, compared and analyzed the advantages and disadvantages of various modeling methods, and improved the defects of 3DMine-Flac3D coupling modeling method. Ding [14] used FLAC 3D to simulate the mining steps of an iron ore stope, and the results showed that when the height of the sublevel reached the corresponding limit, the stope was prone to produce a large number of plastic areas. Wu [15] used FLAC 3D to simulate the underground environment and found that the zonal fracture phenomenon of surrounding rock was largely affected by rock pressure and internal friction Angle. Yu [16] used MIDAS GTS NX for model establishment, then converted model nodes into FLAC 3D identifiable files and brought them into FLAC 3D for simulation, providing a new idea for coupling simulation between FLAC 3D and MIDAS GTS NX. Liang [17] used FLAC 3D to simulate the filling effect of different types of filling bodies and found that bases could enhance the filling effect. Peng [18] found that the failure of backfill was jointly determined by shear and tensile failure, so a new analytical model was proposed to evaluate the stability of backfill. Muhammad [19] et al. established two FLAC 3D models, one for simulating the blasting process and the other for analyzing the stability of filling body, and proposed and discussed the analysis results of filling failure. This method can simulate the stability of blasting impact filling body.

The filling gap changes the integrity of the filling body and weakens the continuity of its structure, thus affecting the strength of the filling body. Once the strength of filling body does not reach the preset value, there will be huge safety risks, which may cause safety accidents such as roof collapse and stope instability. In this paper, the simulation parameters are obtained after triaxial compression test by making the sample of stratified backfill. Then the FLAC 3D and MIDAS GTS NX was used to study the influence of different filling intervals on stope stability.

## Experimental materials and methods

### Physical and chemical properties of tailings

The tailings discharged from Bianjiadayuan lead-zinc-silver ore processing plant were used as filling materials, which were sampled on site and transported to the laboratory for reserve.The tailings of this mine have a pH of 9.The tailings were washed and dried to constant weight to obtain the weight of each grade of tailings. Particle size grading curve (Fig 1) showed the non-uniformity coefficient of tailings particle size is 1.6, indicating good gradation. The median particle size of the tailings is 0.075mm, and the content of fine particles in the tailings is more.

### Preparation of filling body specimen

According to the experimental requirements of the press, the size of the specimen is 50mm in diameter and 100mm in height as shown in Fig 2(A). The concentration of filling slurry used in the experiment is 75% and the ratio of lime to sand is 1:6. It was poured in two phases as shown in Fig 2(B) and 2(C) and the filling intervall time time was 0h,12h,24h,36h and 48h respectively. The curing temperature of the backfill specimen was 20±2°C, and the humidity was above 95% as shown in Fig 2(D).

### Introduction to triaxial compression test

The loading system used in the triaxial compression test (Fig 3) is composed of a universal testing machine and a high confining pressure triaxial visual measuring system. The high

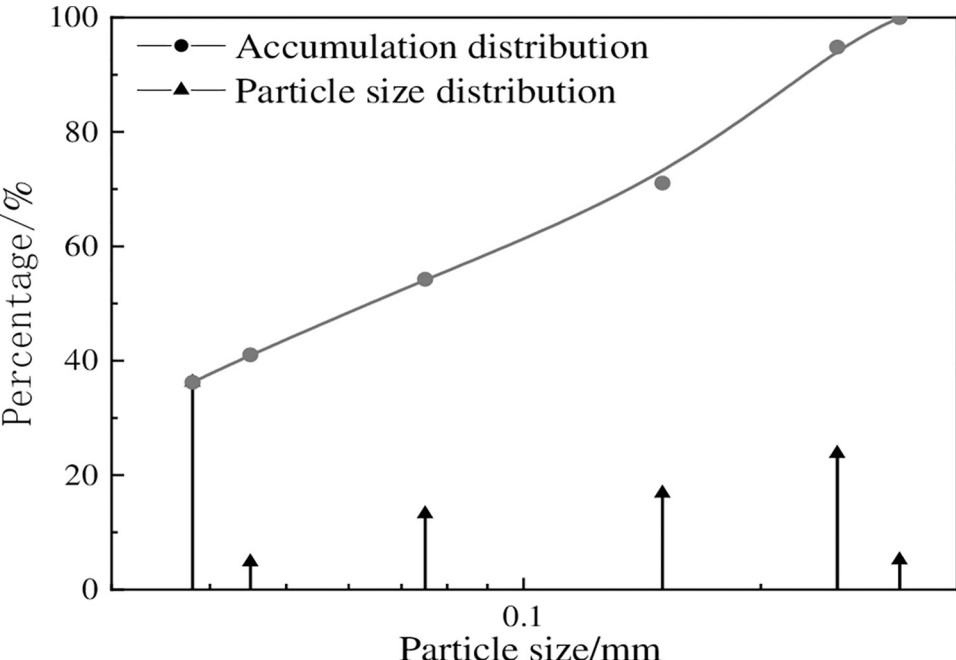

**Fig 1. Grading curve of tailings.**

confining pressure 3D visual measurement system is mainly composed of pressure chamber, camera, pressure sensor and sample image deformation visual system. The main purpose of triaxial test is to provide basic parameters for numerical analysis of stope stability. The experimental target parameters are elastic modulus, internal friction Angle, cohesion, Poisson's ratio.

## Simulation software selection

FLAC 3D can analyze the force and plastic flow of soil, rock and other materials in three-dimensional space. The "mixed discrete method" is used in the simulation process, which is

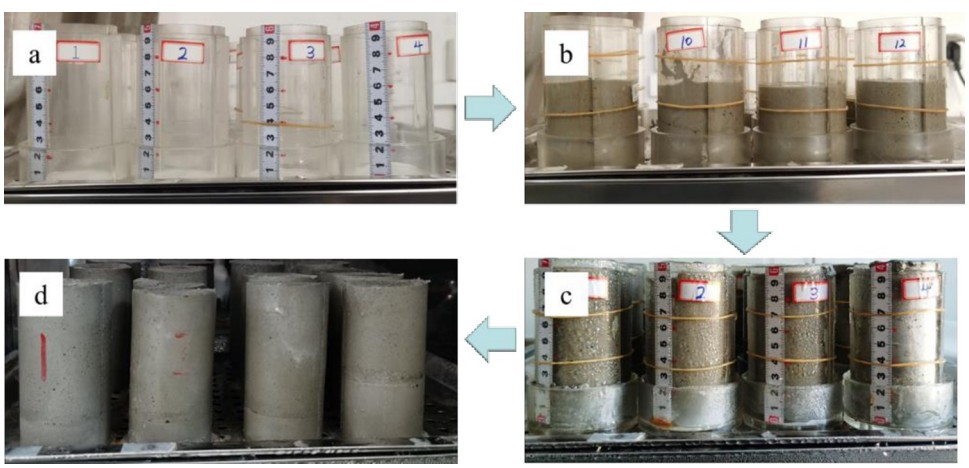

**Fig 2. Specimen making process.**

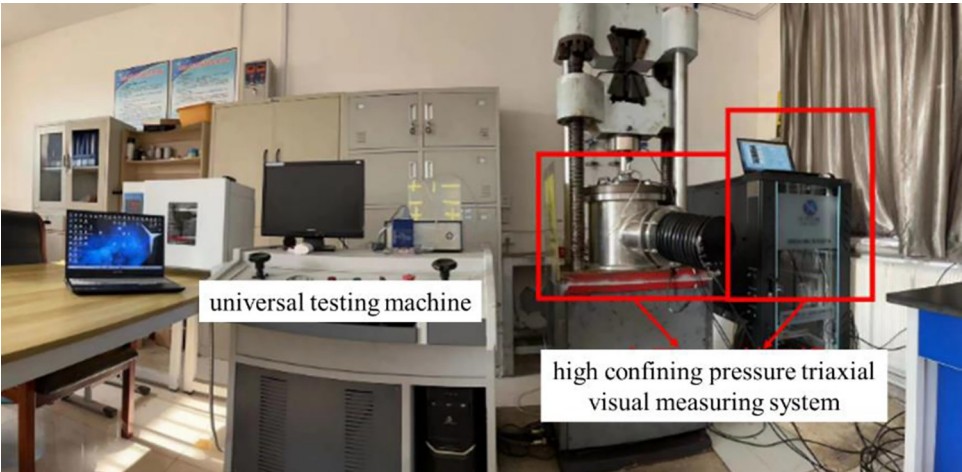

**Fig 3. Triaxial compression experimental system.**

more effective and accurate than the "discrete set method" used by other finite element software.

MIDAS/NTS finite element analysis software has strong advantages in model building and pre-processing such as mesh division. Therefore, MIDAS/NTS finite element analysis software is used to establish the filling stope model. After the model was established, interface tools were used to convert GTS neutral files into FLAC 3D files, and finally imported into FLAC 3D for simulation calculation.

## Model inputs

### Establishment of stope model

Bianjiadayuan lead-zinc-silver mine applied drift stoping and applied the interval mining method.According to the strike of the ore body, the extension is 224m, the dip extension is 105m, and the vertical direction is -571m~-720m. The average dip Angle of the whole ore body in the pillar mining area is 63°. The thickness of the ore body is 15m~30m. According to the field filling body survey, the filling body dimensions on both sides of the pillar are 12m long and 8m high, and the thickness is 24m with the average thickness of the ore body. The strike length of the pillar extension ore body is 4m, the height is 8m, and the average thickness of the ore body is 24m.

MIDAS/NTS finite element analysis software was used to build the model according to stope data. In order to prevent large fluctuations in the data of the simulation results, the model size is set to more than 5 times the mining range of the pillar.

The model is 200m along the dip direction of the ore body, 140m along the strike direction and 200m in the vertical direction, as shown in Fig 4(A). The width interval of filling pillar (along the ore body strike) is 68m~72m, the height interval is 96m~104m, and the average thickness of the ore body is 24m, as shown in Fig 4(B). The width interval of the filling body on one side of the pillar (along the ore body strike) is 56m~68m, the height interval is 96m~104m, and the thickness is 24m. The width interval of the backfill body on the other side (along the ore body strike) is 72m~84m, the height interval is 96m~104m, and the thickness is 24m. The model assumes that the backfill completely contacts the roof and the mined-out areas is excavated smoothly.According to the above data, the filling model was divided into 166,140 units and 130,803 nodes.

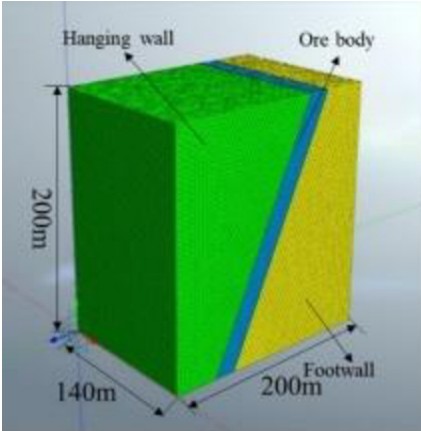
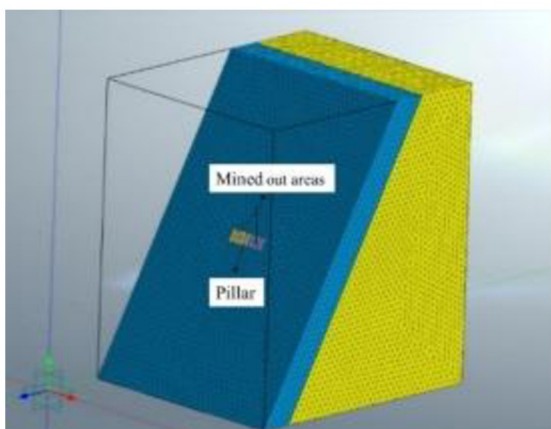

(a) Global model diagram  (b) Remove the hanging plate model diagram

**Fig 4. Underground pillar mining modle.**

## Simulation parameter determination

Triaxial compression tests were carried out on the backfill with different filling intervals to obtain the physical parameters required for stope simulation and ensure that the stope simulation results fit the reality. The experimental instruments and equipment are shown in Fig 5. In the process of experiment, the system photographed the specimen in the pressure chamber in real time, recorded the displacement change of the white recording point on the outer leather sleeve, and calculated the deformation of the specimen. The parameters of filling body at different intervals obtained by the experiment are shown in Table 1 (*FIT* represents filling interval time,*E* represents elastic modulus,*μ* represents poisson's ratio,*C* represents cohesion,*ψ* represents internal friction angle,*TCS* represents triaxial compressive strength, *TS* represents tensile strength, and *ρ* represents density).

Specific parameters of various rocks mass are shown in Table 2.

## Preliminary preparation for numerical simulation

The Molar-Coulomb theory is chosen as the constitutive model.The initial stress was generated by elastoplastic solution with varying strength parameters. In order to prevent the non-convergence of the calculation results in the calculation process, the cohesive force and tensile

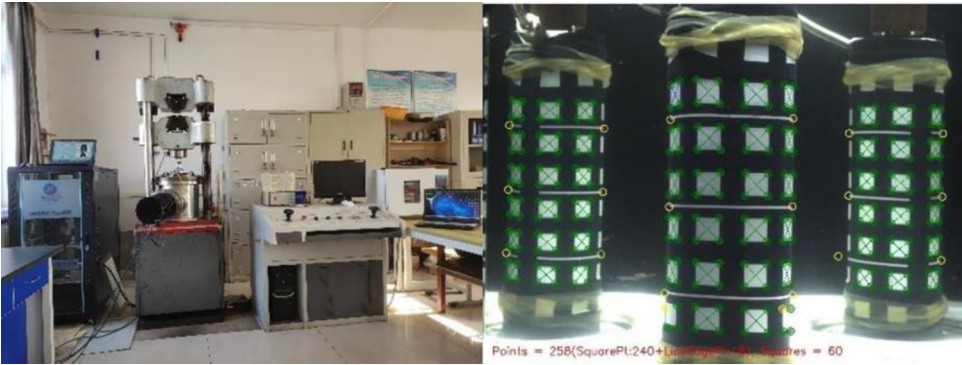

**Fig 5. Experimental equipment.**

**Table 1. Filling body parameters at different intervals.**

| FIT | E/MPa | μ | C/MPa | ψ/(°) | TCS /MPa | TS/MPa | ρ/(kg·m³) |
|-----|-------|---|-------|-------|----------|--------|-----------|
| 0h | 1500 | 0.30 | 0.47 | 39.0 | 4.59 | 0.42 | 1905 |
| 12h | 1460 | 0.31 | 0.43 | 38.1 | 4.31 | 0.40 | 1905 |
| 24h | 1100 | 0.29 | 0.40 | 38.5 | 2.16 | 0.31 | 1905 |
| 36h | 1050 | 0.30 | 0.39 | 37.6 | 1.95 | 0.28 | 1905 |
| 48h | 1000 | 0.31 | 0.38 | 39.4 | 1.81 | 0.27 | 1905 |

strength were set as the maximum value before the formal simulation, and the initial stress generation operation was carried out to resist the damage of the gravitational field to the model. After the initial stress is generated correctly, the cohesive force and tensile strength are set as objective values to conduct formal simulation of stope stability.

## Stope stability analysis

When comparing the influence of different backfill bodies on stope stability, it mainly analyzes from the vertical stress and the displacement near the stope, and compares the filling effect of backfill bodies at different filling intervals.

### Stress distribution analysis

The stress distribution can directly show the stress of stope. The filling body and rock mass have the same physical characteristics, and both belong to compressive resistance and non-tensile resistance. The regularities of stope stress distribution under different filling intervals are shown in Fig 6. The stress of filling body is compared with the compressive and tensile strength of stope roof.

As can be seen from Fig 6(A), when filling backfill with an interval of 0h to support the roof, a small concentration of tensile stress occurs, and the tensile stress value is 0.14881MPa. The stresses in the filling area are all negative, indicating that the filling body area is compressive stress. The compressive stress of the filling body is between 2MPa and 2.5MPa.

As shown in Fig 6(B), the stope vertical tensile stress concentration area of the backfill with a 12h filling interval is larger than that of the backfill with a 0h filling interval, and the tensile stress area extends to the backfill area. The tensile stress was increased to 0.16336MPa. The stresses in the filling area are all negative values between 2MPa and 2.5MPa.

As can be seen from Fig 6(C), compared with the backfill with 0h and 12h filling interval, the vertical tensile stress concentration area of stope increases abruptly, and the tensile stress value also increases to 0.18656MPa. The color gradient in the filling area increased, and the maximum compressive stress in the filling body increased from 2MPa~2.5MPa to 3.0MPa~3.5MPa.

As shown in Fig 6(D), when the interval time of backfill body was 36h, the simulation results were similar to the results of the interval of 24h, and the tensile stress concentration

**Table 2. Rocks mass parameters of filling stope.**

| Rock class | E/MPa | μ | C/MPa | ψ/(°) | TCS /MPa | TS/MPa | ρ/(kg·m³) |
|-----------|-------|---|-------|-------|----------|--------|-----------|
| Ore body | 61400 | 0.26 | 15.2 | 48.7 | 61.1 | 4.9 | 2900 |
| Hanging wall | 79800 | 0.29 | 18.7 | 50.8 | 99.2 | 5.5 | 2812 |
| Footwall | 79700 | 0.29 | 18.7 | 50.8 | 99.2 | 5.5 | 2812 |

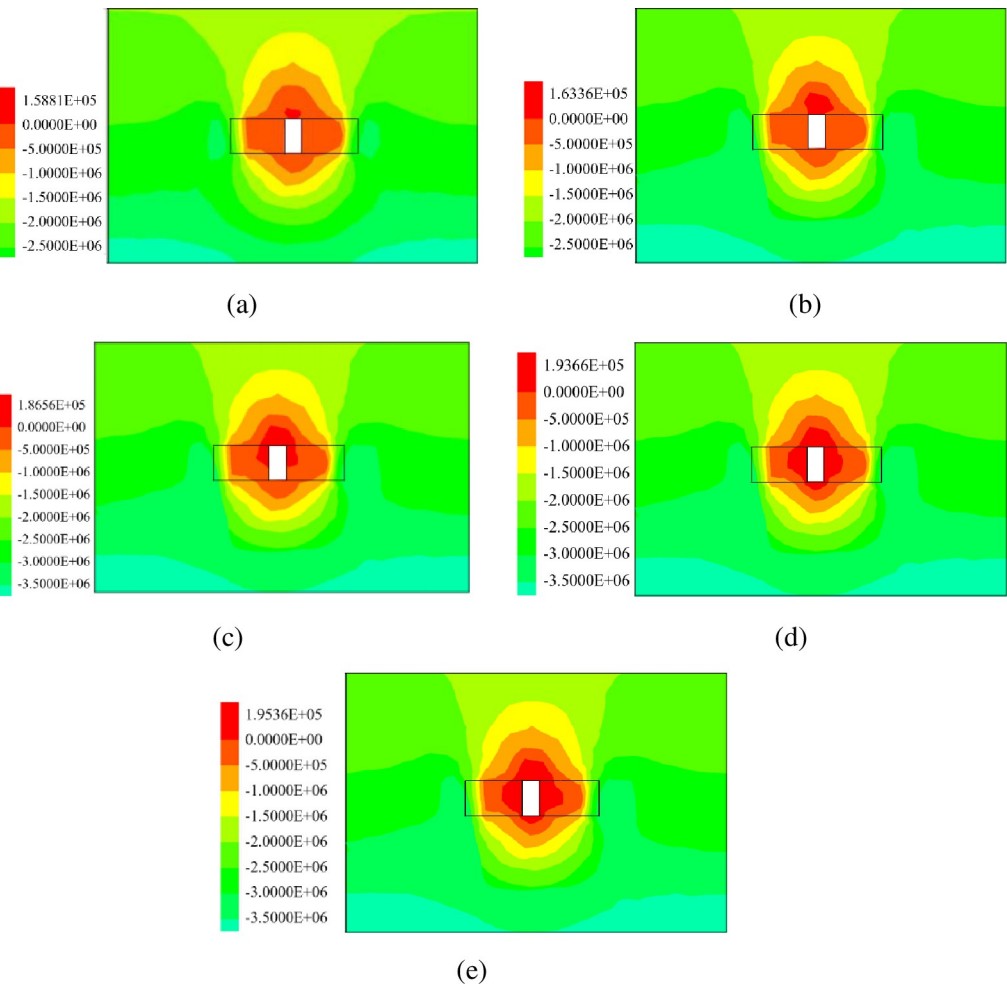

**Fig 6.** Vertical stress nephogram of backfill stope (MPa): (a)0h, (b) 12h, (c) 24h, (d) 36h, (e) 48h.

area of backfill body area gradually expanded, and also increased to 0.19366MPa in numerical aspect. The compressive stress gradient in the filling area is the same as that in 24h, and still within 3.0MPa~3.5MPa.

As shown in Fig 6(E), when the interval time of backfill is 48h, the simulation result is compared with 36h, and the concentrated area of tensile stress in the backfill area gradually tends to be stable, and the tensile stress increases to 0.19536MPa. The compressive stress gradient in the filling area is the same as that in 24h and 36h, which is between 3.0MPa and 3.5MPa.

According to the stress distribution nebulae of stope, it can be found that the roof tensile stress concentration area is small in stope with secondary filling at an interval of 0h~12h. When the secondary filling interval is 12h~24h, the roof tensile stress concentration area of the stope becomes larger rapidly, and the tensile stress concentration area gradually spreads from the roof to the filling body area. The maximum compressive stress gradient in the backfill area increases. When the interval is 24h~48h, the roof tensile stress concentration area has gradually covered the mined-out areas.

Fig 7 shows the maximum stress of stope supported by filling body at different filling intervals. When the filling body interval is from 0h to 12h, the roof tensile stress changes little. At the same time, the compressive stress is between 2MPa~2.5MPa, and the stope stability is

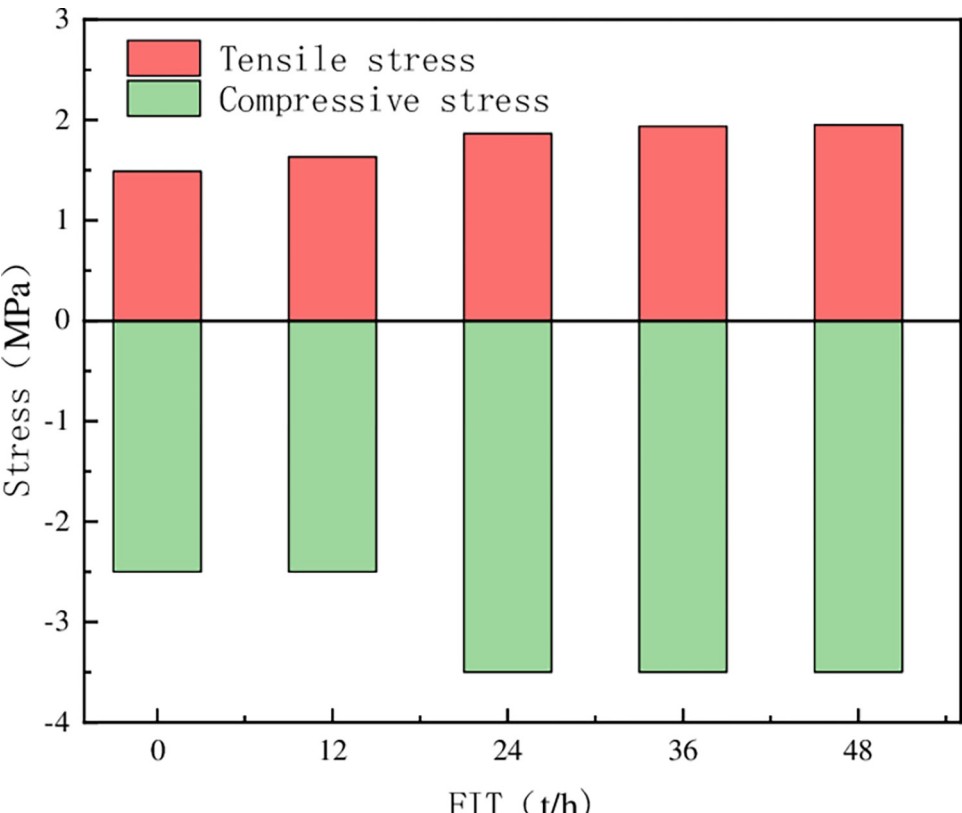

**Fig 7. Variation trend of stope stress with filling interval time.**

good. The maximum compressive stress at 12h~24h is between 3.0MPa~3.5MPa. The maximum tensile stress of stope with an interval of 24h~48h is stable at about 0.19MPa. The maximum compressive stress is also stable between 3.0MPa and 3.5MPa.

Fig 8 shows the comparative relationship between maximum stress and allowable ultimate stress of filling body at different filling intervals. When secondary filling is performed within 0h ~ 12h, the maximum compressive stress on the filling body is less than its ultimate stress, indicating that the stope has good stability. After 12h of secondary filling, the maximum compressive stress of the filling body is greater than its ultimate stress, indicating that the stope is extremely unstable.

According to the results of stress field distribution, the filling interval should not exceed 12h in the secondary filling process of Bijiadayuan lead-zinc-silver mine in Inner Mongolia.

## Stope displacement analysis

Stope roof displacement is an important index to evaluate the filling effect. According to the simulation results of roof displacement by FLAC 3D, the supporting effect of filling body on stope roof at different filling intervals can be analyzed. Stope displacement rule under different filling intervals is shown in Fig 9.

It can be seen from Fig 9(A), when the filling interval is 0h, the roof of the filling stope is somewhat sunk, and the maximum subsidence area is extremely concentrated on the left side of the roof, and the displacement is 7.171mm. The lower part of the stope is uplifted upward. The uplifting area is not large and mainly concentrated on the right side. The uplifting amount is 7.2836mm.

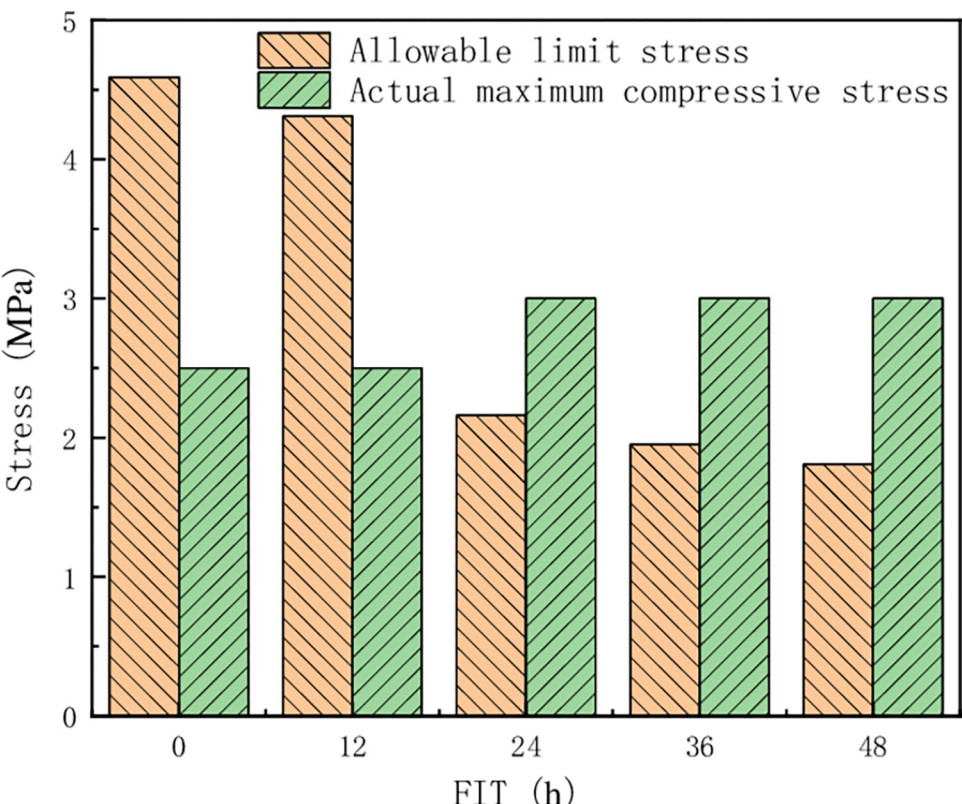

**Fig 8. Comparison diagram of allowable limit stress and maximum stress of filling body at different filling intervals.**

It can be seen from Fig 9(B) that the roof subsidence of the stope with a filling interval of 12h is slightly larger than that of the filling body with a filling interval of 0h, and the maximum subsidence area is slightly increased to 7.311mm. Compared with the uplift area below the stope, the uplift area slightly expands and the uplift displacement increases to 7.461mm.

As can be seen from Fig 9(C), when the filling stope is filled with an interval of 24h, the maximum subsidence area basically covers the roof area, and the displacement rapidly increases to 8.614mm. The uplift displacement under the stope increased to 8.664mm.

As can be seen from Fig 9(D), the displacement of filling stope with filling interval of 36h has a great change compared with the former, and the maximum subsidence area covers all the roof area and gradually extends upward. The displacement of filling stope rapidly increases to 8.844mm. The uplift displacement below the stope increased slightly to 8.852mm.

As can be seen from Fig 9(E), the effect of backfill with filling interval of 48h on supporting stope is not much different from that of 36h, and the maximum subsidence area grows slowly. Displacement of filling stope increased to 8.954mm. The uplift displacement below the stope is slightly increased to 8.976mm.

According to the stress distribution nebulograph of stope, the vertical settlement displacement of the stope roof does not change much during the stope when secondary pouring is carried out at an interval of 0h ~ 12h, increasing from 7.171mm to 7.311mm, and the filling effect is similar.

When the secondary filling interval is 12h ~ 24h, compared with 0h ~ 12h, the roof settlement position decreases abruptly, increasing from 7.311mm to 8.614mm.When the secondary filling interval is 24h ~ 48h, the roof settlement displacement gradually tends to be stable, and

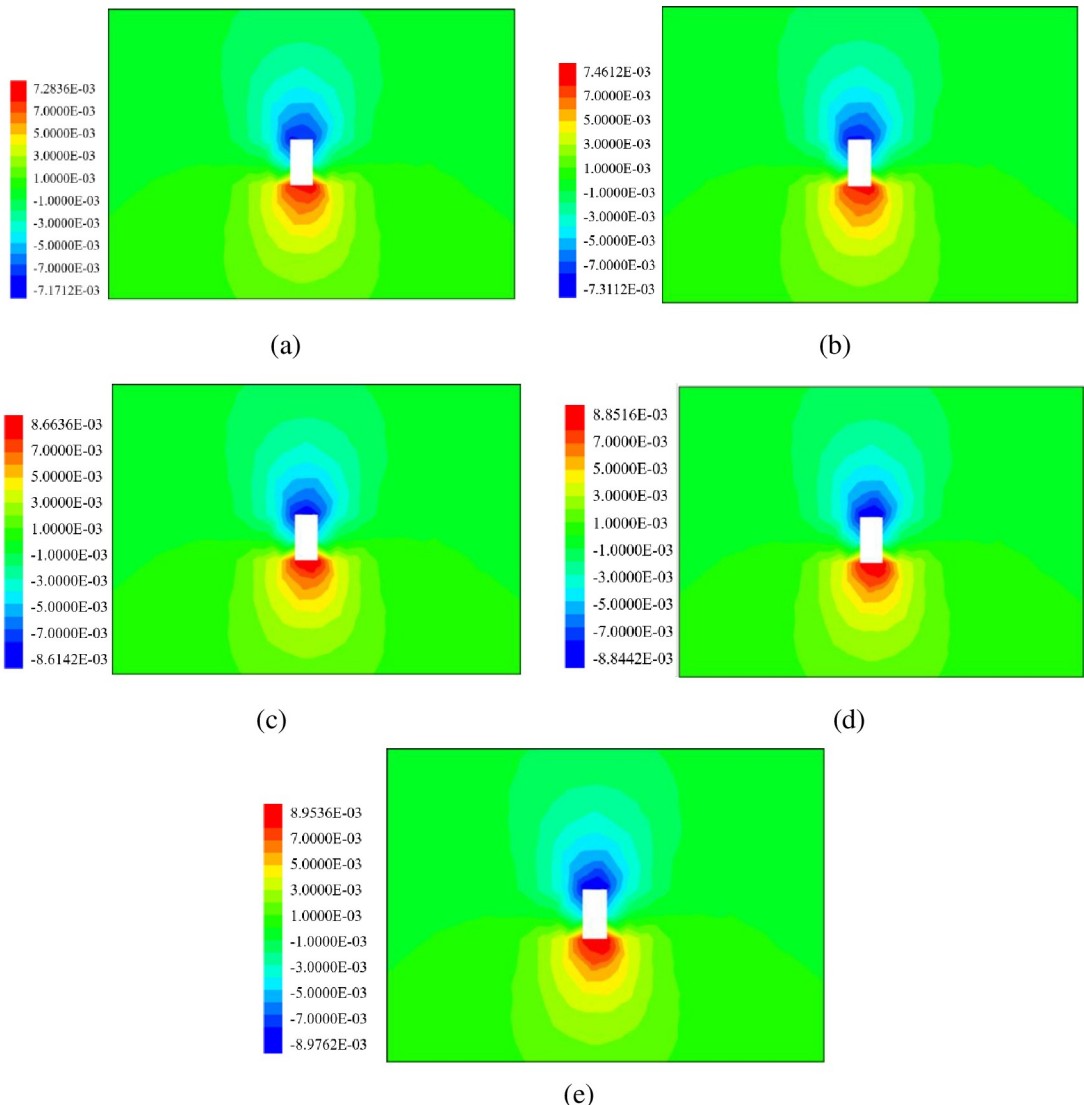

**Fig 9.** Stope displacement diagram of filling body(mm): (a)0h, (b) 12h, (c) 24h, (d) 36h, (e) 48h.

the stope roof displacement supported by the backfill body is 8.614mm, 8.851mm and 8.954mm respectively for 24h, 36h and 48h.

According to the requirements of laboratory test and on-site stope roof displacement control, Roof displacement and the uplift displacement below the stope must not exceed 8mm. When the filling interval is from 0h to 12h, the filling effect of the filling body becomes weaker gradually and meet the requirements. When the interval time is 12h~24h, the filling effect of backfill becomes weak rapidly. When the interval time is 24h, the filling effect becomes weak and the rate slows down again.if the filling interval is longer than 12h, the filling body has poor supporting effect on the stope, which is not conducive to the stability of the stope.

## Conclusion

(1) At 0h~12h, the roof tensile stress concentration area is small and the tensile stress variation is not large. The maximum compressive stress on the backfill body is less than its

ultimate stress, indicating that the stope stability is stable. With the increase of filling interval, the roof tensile stress concentration area of filling stope rapidly expands and covers the mined-out area, indicating that the stope is extremely unstable.

(2) When the filling interval is 0~12h, the vertical settlement displacement of the filling stope roof has little change, increasing from 7.171mm to 7.311mm. When more than 12h, the displacement increases abruptly and then becomes stable.

(3) Through the analysis of the stress and displacement of the filling stope, it is suggested that the interval time in the secondary filling process of Bianjiadayuan lead-zinc-silver mine should not exceed 12h.

## Supporting information

**S1 Data.**
(XLSX)

## Author Contributions

**Conceptualization:** Chunming Ai.

**Data curation:** Chunming Ai.

**Formal analysis:** Chunming Ai.

**Funding acquisition:** Chunming Ai.

**Investigation:** Chunming Ai, Anju Yu.

**Methodology:** Yingbo Wen.

**Project administration:** Chunming Ai.

**Resources:** Yingbo Wen.

**Software:** Anju Yu, Li Zhang, Yingbo Wen.

**Supervision:** Chunming Ai.

**Validation:** Li Zhang.

**Visualization:** Li Zhang.

**Writing – original draft:** Anju Yu, Li Zhang.

**Writing – review & editing:** Anju Yu.

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
