## [Decision Letter · Decision Letter 0]

12 Jul 2023

PONE-D-23-15083Analysis of Influence of Filling Body with Different Filling Intervals on Stope StabilityPLOS ONE

Dear Dr. Yu,

Thank you for submitting your manuscript to PLOS ONE. After careful consideration, we feel that it has merit but does not fully meet PLOS ONE’s publication criteria as it currently stands. Therefore, we invite you to submit a revised version of the manuscript that addresses the points raised during the review process.

We look forward to receiving your revised manuscript.

Kind regards,

Khalil Abdelrazek Khalil, Ph.D.

Academic Editor

PLOS ONE

Reviewers' comments:

Reviewer's Responses to Questions

**Comments to the Author**

1. Is the manuscript technically sound, and do the data support the conclusions?

Reviewer #1: Yes

Reviewer #2: Partly

Reviewer #3: Partly

2. Has the statistical analysis been performed appropriately and rigorously? 

Reviewer #1: Yes

Reviewer #2: Yes

Reviewer #3: Yes

3. Have the authors made all data underlying the findings in their manuscript fully available?

Reviewer #1: Yes

Reviewer #2: Yes

Reviewer #3: Yes

4. Is the manuscript presented in an intelligible fashion and written in standard English?

Reviewer #1: Yes

Reviewer #2: Yes

Reviewer #3: No

5. Review Comments to the Author

Reviewer #1: 1、This paper discusses the effect of filling interval time on stope stability, and gives a reference filling interval time, which should be reflected in the conclusion.

2、This paper introduces the specimen making model, but does not introduce the stratified specimen making process. It is suggested to briefly introduce the production method of the specimen of the stratified backfill.

3、Figure 3 should be marked in English.

4、The acceptable range of stress and displacement after filling the backfill to the goaf should be described briefly.

5、You should list the Idealized condition when creating the model.

Reviewer #2: In view of the research content and methods of this paper, I have to reject the manuscript for the following reasons.

1.The research content is too general, lack of novelty, in addition to the lack of theoretical research, the content of the paper is insufficient.

2.The physical test is not detailed, the test process is not clear, and the physical test part does not explain the test results.

3.There is no conclusion in the manuscript, and what kind of conclusion is drawn through the research is not explained.

4.4.Paper format is not standardized.

Reviewer #3: 1. The word “goaf” is not usually used in a mining manuscript, it would be better to use words “mined-out areas”.

2. Does “fractional filling” means “stratified backfill”?

3. In Introduction, in my opinion, backfill capacity is no longer a limit factor for the choice of mining method, but the rock mass characteristics is.

4. The information of x-coordinate and y-coordinate titles of Fig. 2.1 is insufficient.

5. The legend in Fig. 2.3 is in Chinese, which is inappropriate. Also, the picture resolution in not good enough.

6. In the triaxial test, authors should briefly introduce how to make the layered cemented backfill specimen and what production conditions are taken into account.

7. In Section 2.2 and 2.3, authors made some backfill samples in size of 50*100mm, it’s a typical size for a triaxial UCS test. However, most industrial codes or standards declare that the cubic specimen 70.7mm*70.7mm*70.7mm should be used for strength test. Please confirm the sample reliability. Also, the test steps should be described in detail.

8. In Section 3.1, the filling situation of Bianjiadayuan lead-zinc-silver ore is taken into account when the filling model is established. It would be better to introduce the onsite geological condition and filling situation of mine, and combine with the model.

9. When the model is established, there are some conditions that are difficult to be completely simulated, such as the flatness of the gob and the top-connecting effect of the backfill, and these neglected conditions should be expressed in the paper.

10. Useful information such as model size, direction, rock mass type, should be added to Fig. 3.1. The current picture is too simple to support the analysis.

11. In line 5 and Table 3.2 of Section 3.2, all variables should be italic. some variable names are missing. Pls also note the capital and lower letter. (e.g. friction Angle?) The surrounding rock mass should be hanging wall & footwall.

12. Line 1 of Section 3.3, “mol Coulomb yield theory”??

13. In numerical simulations, physical and mechanical parameters usually refer to rock mass instead of rock, however, Table 3.2 clearly shows only rock mechanics.

14. It’s opinionated to determine that “there are on bad cracks in the rock mass” since the invisibility within the rock mass. And, does “bad crack” mean geological structure?

15. Authors are encouraged to detailedly illustrate the term “ filling intervals” with proper languages and figures. The term is very important in the manuscript.

16. In Section 4.1, what mining method is used? what’s the relationship and arrangement of mining stope and its neighbouring stopes?

17. The typeface in Fig. 4.2 should not be in Chinese. The information of x-coordinate and y-coordinate titles is too simple to provide useful conclusions.

18. Though Mohr-Coulomb criterion is widely used in underground mining, authors are encouraged to set a factor of safety to assess the stope stability besides allowable limit stress.

19. In Section 4.2, it’s also necessary to state the criterion of roof failure. The roof displacement in 7~8 mm shows little difference. Authors should provide the failure threshold value.

20. Besides vertical displacement and principal stress, the distribution of plastic zone is also very important for stope stability. Authors should add these simulated results.

21. Pay attention to article formatting, such as first line indentation, chart position.

6. PLOS authors have the option to publish the peer review history of their article (what does this mean?). If published, this will include your full peer review and any attached files.

Reviewer #1: No

Reviewer #2: No

Reviewer #3: No

---

## [Author Response · Author response to Decision Letter 0]

25 Jul 2023

《Analysis of Influence of Filling Body with Different Filling Intervals on Stope Stability》Manuscript revision comments reply

Dear editors and experts of Journal of China University of Plos One:

Hello! Many thanks to the editors and experts of the journal for their detailed review and pertinent opinions. Now the opinions have been revised in the paper, and detailed explanations are made here, hoping that the editors and experts can approve.I wish you good health and good luck!

Reply to Expert I: 

1、This paper discusses the effect of filling interval time on stope stability, and gives a reference filling interval time, which should be reflected in the conclusion.

Reply: A summary had been added to chapter 5 of the original text. The changes of tensile stress, compressive stress and displacement of the roof in the study were summarized as follows:

 (1) At 0h~12h, the roof tensile stress concentration area is small, the tensile stress variation is not large. The maximum compressive stress on the backfill body is less than its ultimate stress, indicating that there is no plastic change in the filling area at this time and the stope stability is stable. With the increase of filling interval, the roof tensile stress concentration area of filling stope rapidly expands and covers the mined-out area, plastic deformation occurs in the filling body area, and the stope is extremely unstable.

 (2) When secondary pouring is carried out at intervals of 0h ~ 12h, the vertical settlement displacement of the stope roof has little change during the stoping process. With the increasing of the interval time, the displacement increases abruptly and finally becomes stable.

 (3) Through the analysis of the stress and displacement of the filling stope, it is suggested that the interval time in the secondary filling process of Bijiadayuan Pb-Zn-silver mine should not exceed 12h.

2、This paper introduced the specimen making model, but does not introduce the stratified specimen making process. It is suggested to briefly introduce the production method of the specimen of the stratified backfill.

Reply：The production process of filling body specimen has been added in 2.2. (Preparation of filling body specimen), The production process and production conditions of test pieces are briefly introduced as follows:

According to the experimental requirements of the press, the size of the specimen is 50mm in diameter and 100mm in height as shown in Figure 2(a). the concentration of filling slurry used in the experiment is 75% and the ratio of lime to sand is 1:6. It was poured in two phases as shown in Figure 2(b)(c).The curing temperature of the backfill specimen was 20±2℃, and the humidity was above 95% as shown in Figure 2(d).

3、Figure 3 should be marked in English.

Reply：The image has been marked in English.

4、The acceptable range of stress and displacement after filling the backfill to the goaf should be described briefly.

Reply：The displacement threshold was added to 4.2 .

During the triaxial test, the specimens of the backfill were compressed 7-7.5mm, and no obvious damage occurred. When the specimen exceeds 7.5mm, the specimen crack develops gradually. According to the actual requirements of Bianjiadayuan lead-zinc-silver ore, the displacement should not exceed 8mm. Therefore, this paper sets the displacement threshold as 8mm, and if it exceeds 8mm, it is regarded as an unsatisfactory state. This study reflects the variation of tensile stress coverage and the maximum stress of stope supported by backfill, and is compared with the allowable limit stress, which can illustrate the stability of stope. Add content as follows:

 According to the requirements of laboratory test and on-site stope roof displacement control, stope roof displacement should not exceed 8mm.

5、You should list the Idealized condition when creating the model

Reply：The conditions that affect the simulation results, such as topping rate and flatness, are listed as ideal conditions.Ideal conditions added to 3.1.Add content as follows:

The model assumes that the backfill completely contacts the roof and the mined-out areas is excavated smoothly.

Reply to ExpertⅡ: 

1、The research content is too general, lack of novelty, in addition to the lack of theoretical research, the content of the paper is insufficient.

Reply：A large number of scholars simulated the stability of stope through Flac3D software.In this paper, the displacement and stress distribution characteristics of stope under different stope filling intervals were studied, and the changing rules of stope stability under different intervals were obtained.

2、The physical test is not detailed, the test process is not clear, and the physical test part does not explain the test results.

Reply：The process of physical testing in this paper is only to obtain simulation parameters, not the real content to be analyzed. The simulation parameters obtained are shown in Table 1.

Table 1. Filling body parameters at different intervals

FIT E/MPa μ C/MPa ψ/(°) TCS

/MPa TS/MPa ρ/(kg·m3)

0h 1500 0.30 0.47 39.0 4.59 0.42 1905

12h 1460 0.31 0.43 38.1 4.31 0.40 1905

24h 1100 0.29 0.40 38.5 2.16 0.31 1905

36h 1050 0.30 0.39 37.6 1.95 0.28 1905

48h 1000 0.31 0.38 39.4 1.81 0.27 1905

3、There is no conclusion in the manuscript, and what kind of conclusion is drawn through the research is not explained.

Reply：A summary had been added to chapter 5 of the original text. The changes of tensile stress, compressive stress and displacement of the roof in the study were summarized.

4、Paper format is not standardized.

Reply：The paper format has been modified to conform to 《Plos one》 journal format.

Reply to Expert Ⅲ: 

1、The word “goaf” is not usually used in a mining manuscript, it would be better to use words “mined-out areas”.

Reply：After reviewing the literature and expert guidance, the "goaf" in the article has been replaced with "mined out areas".

2、Does “fractional filling” means “stratified backfill”?

Reply：The statement of stratified backfill is incorrect.all “fractional filling” in the article have been changed to “stratified backfill”.

3、In Introduction, in my opinion, backfill capacity is no longer a limit factor for the choice of mining method, but the rock mass characteristics is.

Reply: The introduction is inaccurate and has been revised.

4、The information of x-coordinate and y-coordinate titles of Fig. 2.1 is insufficient.

Reply：The information of x-coordinate and y-coordinate titles of Fig. 2.1 has been modified. The modified image is as follows:

5、The legend in Fig. 2.3 is in Chinese, which is inappropriate. Also, the picture resolution in not good enough.

Reply：The pictures have been annotated in English. The modified image is as follows：

6、In the triaxial test, authors should briefly introduce how to make the layered cemented backfill specimen and what production conditions are taken into account.

Reply：The production process of filling body specimen has been added in 2.2. (Preparation of filling body specimen), The production process and production conditions of test pieces are briefly introduced as follows:

According to the experimental requirements of the press, the size of the specimen is 50mm in diameter and 100mm in height as shown in Figure 2(a). the concentration of filling slurry used in the experiment is 75% and the ratio of lime to sand is 1:6. It was poured in two phases as shown in Figure 2(b)(c).The curing temperature of the backfill specimen was 20±2℃, and the humidity was above 95% as shown in Figure 2(d).

7、In Section 2.2 and 2.3, authors made some backfill samples in size of 50*100mm, it’s a typical size for a triaxial UCS test. However, most industrial codes or standards declare that the cubic specimen 70.7mm*70.7mm*70.7mm should be used for strength test. Please confirm the sample reliability. Also, the test steps should be described in detail.

Reply：The specimens in this paper are made according to the requirements of the pseudo-triaxial test and have reliability. The high confining pressure 3D visual measurement system is mainly composed of pressure chamber, camera, pressure sensor and sample image deformation visual system. In the process of experiment, the system photographed the specimen in the pressure chamber in real time, recorded the displacement change of the white recording point on the outer leather sleeve, and calculated the deformation of the specimen. This content has been added to 2.3.

8、In Section 3.1, the filling situation of Bianjiadayuan lead-zinc-silver ore is taken into account when the filling model is established. It would be better to introduce the onsite geological condition and filling situation of mine, and combine with the model.

Reply：The onsite geological condition and filling situation of mine has been added to 3.1. Add content as follows:

Bianjiadayuan lead-zinc-silver mine applied drift stoping and applied the interval mining method.According to the strike of the ore body, the extension is 224m, the dip extension is 105m, and the vertical direction is -571m~-720m. The average dip Angle of the whole ore body in the pillar mining area is 63°. The thickness of the ore body is 15m~30m. According to the field filling body survey, the filling body dimensions on both sides of the pillar are 12m long and 8m high, and the thickness is 24m with the average thickness of the ore body. The strike length of the pillar extension ore body is 4m, the height is 8m, and the average thickness of the ore body is 24m.

9、When the model is established, there are some conditions that are difficult to be completely simulated, such as the flatness of the gob and the top-connecting effect of the backfill, and these neglected conditions should be expressed in the paper.

Reply：The Ideal condition has been added to 3.1. Add content as follows:

The model assumes that the backfill completely contacts the roof and the mined-out areas is excavated smoothly.

10、Useful information such as model size, direction, rock mass type, should be added to Fig. 3.1. The current picture is too simple to support the analysis.

Reply：The model size、rock mass type has been added to Fig. 3.1. What cannot be marked is written.

The model is 200m along the dip direction of the ore body, 140m along the strike direction and 200m in the vertical direction, as shown in Figure 4 (a). The width interval of filling pillar (along the ore body strike) is 68m~72m, the height interval is 96m~104m, and the average thickness of the ore body is 24m, as shown in Figure 4 (b). The width interval of the filling body on one side of the pillar (along the ore body strike) is 56m~68m, the height interval is 96m~104m, and the thickness is 24m. The width interval of the backfill body on the other side (along the ore body strike) is 72m~84m, the height interval is 96m~104m, and the thickness is 24m.

11、In line 5 and Table 3.2 of Section 3.2, all variables should be italic. some variable names are missing. Pls also note the capital and lower letter. (e.g. friction Angle?) The surrounding rock mass should be hanging wall & footwall.

Reply: All variables have been italicized. The description of the variable FIT is completed. The expression in the article has been changed to hanging wall & footwall. The changes are as follows:

(FIT represents filling interval time,E represents elastic modulus,μ represents poisson's ratio,C represents cohesion,ψ represents internal friction angle,TCS represents triaxial compressive strength, TS represents tensile strength, and ρ represents density).

12、Line 1 of Section 3.3, “mol Coulomb yield theory”?

Reply：Statement a is inaccurate and has been deleted. The changed content is as follows：

The Molar-Coulomb theory is chosen as the constitutive model.

13、In numerical simulations, physical and mechanical parameters usually refer to rock mass instead of rock, however, Table 3.2 clearly shows only rock mechanics.

Reply：Parameter description is wrong. the actual reference is the rock mass parameter. The changes are as follows:

Specific parameters of various rocks mass are shown in Table 2.

Table 2. Rocks mass parameters of filling stope

Rock class E/MPa μ C/MPa ψ/(°) TCS

/MPa TS/MPa ρ/(kg·m3)

Ore body 61400 0.26 15.2 48.7 61.1 4.9 2900

Hanging wall 79800 0.29 18.7 50.8 99.2 5.5 2812

Footwall 79700 0.29 18.7 50.8 99.2 5.5 2812

14、It’s opinionated to determine that “there are on bad cracks in the rock mass” since the invisibility within the rock mass. And, does “bad crack” mean geological structure?

Reply：The content is incorrectly stated. The content has been changed.

15、Authors are encouraged to detailedly illustrate the term “ filling intervals” with proper languages and figures. The term is very important in the manuscript.

Reply：The filling interval has been briefly introduced, which is reflected in 2.2. The changes are as follows:

It was poured in two phases as shown in Figure 2(b),2(c) and the filling intervall time time was 0h,12h,24h,36h and 48h respectively. 

16、In Section 4.1, what mining method is used? what’s the relationship and arrangement of mining stope and its neighbouring stopes?

Reply：The mining method and the relationship and arrangement of mining stope and its neighbouring stopes have been added in 3.1, which are described as follows：

Bianjiadayuan lead-zinc-silver mine applied drift stoping and applied the interval mining method.

17、The typeface in Fig. 4.2 should not be in Chinese. The information of x-coordinate and y-coordinate titles is too simple to provide useful conclusions.

Reply：The content in the figure has been expressed in text, and it is indeed superfluous in the article, so it is deleted.

18、Though Mohr-Coulomb criterion is widely used in underground mining, authors are encouraged to set a factor of safety to assess the stope stability besides allowable limit stress.

Reply: The comparison between the actual maximum compressive stress and the allowable limit stress is used to evaluate the stability of stope. The stability of stope is not evaluated from the Angle of safety factor.

19、In Section 4.2, it’s also necessary to state the criterion of roof failure. The roof displacement in 7~8 mm shows little difference. Authors should provide the failure threshold value.

Reply: When the triaxial compression test was carried out, no obvious damage occurred when the compression displacement was 7mm-8mm. When the compression displacement exceeds 8mm, the specimen will appear serious spalling and crack growth. Combined with the requirements of the mine on the displacement of the roof, the threshold is set at 8mm. The content has been added to 4.2 and is as follows.

According to the requirements of laboratory test and on-site stope roof displacement control, stope roof displacement should not exceed 8mm.

20、Besides vertical displacement and principal stress, the distribution of plastic zone is also very important for stope stability. Authors should add these simulated results.

Reply: In this paper, when studying the influence of filling interval on stope stability, the plastic zone of stope surrounding rock is small and the difference is not obvious, so the main content is not expounded. Plastic zone of filling body changes little with different filling intervals. By comparing the maximum compressive stress with the ultimate compressive stress, the variation of the plastic zone in the backfill area is simply described.

21、Pay attention to article formatting, such as first line indentation, chart position.

Reply: Changes have been made to the first line of indentation a

---

## [Decision Letter · Decision Letter 1]

2 Oct 2023

PONE-D-23-15083R1Analysis of Influence of Filling Body with Different Filling Intervals on Stope StabilityPLOS ONE

Dear Dr. Yu,

Thank you for submitting your manuscript to PLOS ONE. After careful consideration, we feel that it has merit but does not fully meet PLOS ONE’s publication criteria as it currently stands. Therefore, we invite you to submit a revised version of the manuscript that addresses the points raised during the review process.

We look forward to receiving your revised manuscript.

Kind regards,

Khalil Abdelrazek Khalil, Ph.D.

Academic Editor

PLOS ONE

Journal Requirements:

Reviewers' comments:

Reviewer's Responses to Questions

**Comments to the Author**

1. If the authors have adequately addressed your comments raised in a previous round of review and you feel that this manuscript is now acceptable for publication, you may indicate that here to bypass the “Comments to the Author” section, enter your conflict of interest statement in the “Confidential to Editor” section, and submit your "Accept" recommendation.

Reviewer #1: All comments have been addressed

Reviewer #3: All comments have been addressed

2. Is the manuscript technically sound, and do the data support the conclusions?

Reviewer #1: Yes

Reviewer #3: Partly

3. Has the statistical analysis been performed appropriately and rigorously? 

Reviewer #1: Yes

Reviewer #3: Yes

4. Have the authors made all data underlying the findings in their manuscript fully available?

Reviewer #1: Yes

Reviewer #3: Yes

5. Is the manuscript presented in an intelligible fashion and written in standard English?

Reviewer #1: Yes

Reviewer #3: No

6. Review Comments to the Author

Reviewer #1: After reading the article, the author has made modifications according to the requirements and met the acceptance criteria for the paper.

Reviewer #3: Authors revised the manuscript according to reviewer’s comments, in my opinion, they provided key information to better describe their work. The revision is reliable. Some comments as given as follows:

1. It’s a mistake to write “Dear editors and experts of Journal of China University of Plos One”, this silly mistake should be avoided.

2. The first letter of the first word in the Abstract should be capital.

3. One sentence is encouraged to add to the end of abstract to state the theoretical or engineering significance of the work.

4. English should be polished throughout the manuscript.

5. In introduction, “at home and abroad” is better replaced by “worldwide”.

6. The objective, methods and work procedure should be stated in the Introduction.

7. “3.3 Preliminary preparation”, “3.3” should be removed.

8. Fig. 6 lacks of stress unit, N? kN? Pls also check all similar figures.

9. Fig. 6 &9 do not illustrate the stability of neighboring stopes, only filled stope (white rectangle) is showed. It’s more important to know the stability of neighboring stopes that about to mining. These two picture should be further edited to provide more detailed information.

10. To better introduce Fig. 6 &9, stress concentration zone is encouraged to be included.

11. In Conclusion 1, the backfill suffers lower compressive stress that its ultimate stress does not mean the state of plastic zone distribution. Few backfill body is damaged by compressive stress compare to tensile stress or shear stress.

7. PLOS authors have the option to publish the peer review history of their article (what does this mean?). If published, this will include your full peer review and any attached files.

Reviewer #1: No

Reviewer #3: No

---

## [Author Response · Author response to Decision Letter 1]

9 Oct 2023

《Analysis of Influence of Filling Body with Different Filling Intervals on Stope Stability》Manuscript revision comments reply

Dear editors and experts of Plos One:

Hello! Many thanks to the editors and experts of the journal for their detailed review and pertinent opinions. Now the opinions have been revised in the paper, and detailed explanations are made here, hoping that the editors and experts can approve.I wish you good health and good luck!

Reply to Expert Ⅲ: 

1、It’s a mistake to write “Dear editors and experts of Journal of China University of Plos One”, this silly mistake should be avoided.

Reply: Changes have been made as requested.

2、Line 21-22: The first letter of the first word in the Abstract should be capital..

Reply：Changes have been made as requested.

The stratified backfill will lead to the reduction of the strength of the filling body, thus increasing the safety risks of the stope.

3、One sentence is encouraged to add to the end of abstract to state the theoretical or engineering significance of the work.

Reply：Relevant content has been added to the original manuscript. Add as follows:

The research results have significant reference value for improving the stope stability and selecting a reasonable interval time for stratified backfill. 

4、English should be polished throughout the manuscript.

Reply：The author has polished the whole text to reduce grammatical and spelling errors.

5、In introduction, “at home and abroad” is better replaced by “worldwide”.

Reply：Changes have been made as requested.

To solve the stability problem of filling stope, numerical simulation method is widely used worldwide, which can directly reflect the change characteristics of displacement of detection points.

6、The objective, methods and work procedure should be stated in the Introduction

Reply：Relevant content has been added to the original manuscript. Add as follows:

In this paper, the simulation parameters are obtained after triaxial compression test by making the sample of stratified backfill. Then the FLAC 3D and MIDAS GTS NX was used to study the influence of different filling intervals on stope stability.

The objective: Therefore, this paper adopts numerical simulation method to study the influence of different filling intervals on stope stability.

Methods and work procedure: It is explained in detail in “Experimental materials and methods”

7、Line 83: “3.3 Preliminary preparation”, “3.3” should be removed.

Reply：3.3 has been removed.

8、Fig. 6 lacks of stress unit, N? kN? Pls also check all similar figures.

Reply: When the picture was exported by software, no suitable unit was provided, so the author added the unit to the illustration. Add as follows:

Fig 6. Vertical stress nephogram of backfill stope (MPa):

Fig 9. Stope displacement diagram of filling body (mm)

9、Fig. 6 &9 do not illustrate the stability of neighboring stopes, only filled stope (white rectangle) is showed. It’s more important to know the stability of neighboring stopes that about to mining. These two picture should be further edited to provide more detailed information.

Reply: This paper mainly studies the effect of filling interval time on stope stability. According to the simulation results, different filling intervals will affect the strength of the filling body. The shorter the filling interval, the higher the strength of the filling body, that is, the higher the stope stability, which provides a reasonable selection range for the filling interval time. The study on the stability of adjacent stope is not in the focus of this paper, so it is not included in the article. 

10、To better introduce Fig. 6 &9, stress concentration zone is encouraged to be included.

Reply: The author believes that the image exported by the software can also clearly represent the stress concentration area, which may not be clear enough if it is enlarged.

11、In Conclusion 1, the backfill suffers lower compressive stress that its ultimate stress does not mean the state of plastic zone distribution. Few backfill body is damaged by compressive stress compare to tensile stress or shear stress.

Reply: Changes have been made to the expression in the text .Because the tensile stress of filling body changes little, the main contrast between compressive stress and ultimate stress, that is, the lower the compressive stress than the ultimate stress, the more stable the stope. The change of plastic zone is not considered and mentioned in this paper:

At 0h~12h, the roof tensile stress concentration area is small and the tensile stress variation is not large. The maximum compressive stress on the backfill body is less than its ultimate stress, indicating that the stope stability is stable. With the increase of filling interval, the roof tensile stress concentration area of filling stope rapidly expands and covers the mined-out area, indicating that the stope is extremely unstable.

---

## [Decision Letter · Decision Letter 2]

16 Oct 2023

Analysis of Influence of Filling Body with Different Filling Intervals on Stope Stability

PONE-D-23-15083R2

Dear Dr. Yu,

We’re pleased to inform you that your manuscript has been judged scientifically suitable for publication and will be formally accepted for publication once it meets all outstanding technical requirements.

Kind regards,

Khalil Abdelrazek Khalil, Ph.D.

Academic Editor

PLOS ONE

Additional Editor Comments (optional):

Reviewers' comments:

Reviewer's Responses to Questions

**Comments to the Author**

1. If the authors have adequately addressed your comments raised in a previous round of review and you feel that this manuscript is now acceptable for publication, you may indicate that here to bypass the “Comments to the Author” section, enter your conflict of interest statement in the “Confidential to Editor” section, and submit your "Accept" recommendation.

Reviewer #3: All comments have been addressed

2. Is the manuscript technically sound, and do the data support the conclusions?

Reviewer #3: Yes

3. Has the statistical analysis been performed appropriately and rigorously? 

Reviewer #3: Yes

4. Have the authors made all data underlying the findings in their manuscript fully available?

Reviewer #3: Yes

5. Is the manuscript presented in an intelligible fashion and written in standard English?

Reviewer #3: Yes

6. Review Comments to the Author

Reviewer #3: This revised version meets the review requirement. The following suggestion may be helpful to improve the manuscript. I would suggest publication without further review.

1, In abstract, “filling body pressure stress” should be “ filling body suffered pressure stress”.

2, In Abstract, “ ...have significant reference value…” is somewhat exaggerated. It’s suggested to use “..is beneficial to improve … and select …”.

3, In Introduction, “the height … is too large”? large height?

4, In Introduction, “friction Angle”?

5, “Fig 4. Underground pillar mining modle”, modle? Global model diagram?

7. PLOS authors have the option to publish the peer review history of their article (what does this mean?). If published, this will include your full peer review and any attached files.

Reviewer #3: No

---

## [Editor Report · Acceptance letter]

19 Oct 2023

PONE-D-23-15083R2 

Analysis of Influence of Filling Body with Different Filling Intervals on Stope Stability 

Dear Dr. Yu:

I'm pleased to inform you that your manuscript has been deemed suitable for publication in PLOS ONE. Congratulations! Your manuscript is now with our production department. 

Kind regards, 

on behalf of

Dr. Khalil Abdelrazek Khalil 

Academic Editor

PLOS ONE